# Is Bitcoin Still a King? Relationships between Prices, Volatility and Liquidity of Cryptocurrencies during the Pandemic

**DOI:** 10.3390/e23111386

**Published:** 2021-10-22

**Authors:** Barbara Będowska-Sójka, Agata Kliber, Aleksandra Rutkowska

**Affiliations:** 1Department of Econometrics, Poznan University of Economics and Business, 61-875 Poznan, Poland; barbara.bedowska-sojka@ue.poznan.pl; 2Department of Applied Mathematics, Poznan University of Economics and Business, 61-875 Poznan, Poland; aleksandra.rutkowska@ue.poznan.pl

**Keywords:** cryptocurrencies, mutual information, transfer entropy, dynamic time warping

## Abstract

We try to establish the commonalities and leadership in the cryptocurrency markets by examining the mutual information and lead-lag relationships between Bitcoin and other cryptocurrencies from January 2019 to June 2021. We examine the transfer entropy between volatility and liquidity of seven highly capitalized cryptocurrencies in order to determine the potential direction of information flow. We find that cryptocurrencies are strongly interrelated in returns and volatility but less in liquidity. We show that smaller and younger cryptocurrencies (such as Ripple’s XRP or Litecoin) have started to affect the returns of Bitcoin since the beginning of the pandemic. Regarding liquidity, the results of the dynamic time warping algorithm also suggest that the position of Monero has increased. Those outcomes suggest the gradual increase in the role of privacy-oriented cryptocurrencies.

## 1. Introduction

Bitcoin is the most noticeable cryptocurrency in the fast-growing market [1]. However, because the number of currencies has been rapidly growing and investors face different investment opportunities, its dominance is disputable. This paper aims to analyse the links between leading cryptocurrencies. These links are measured by the amount of information shared and transmitted before and during the pandemic. We also verify possible lead-lag relationships within the sample. We study seven cryptocurrencies of the highest market capitalization and a relatively long history of market quotations. The coronavirus pandemic and the resulting unprecedented crisis has affected the entire investment community, and many assets and commodities significantly dropped in value. We focus on cryptocurrencies that—on the contrary—experienced an increase in their value (at the beginning of 2020, Bitcoin price oscillated around 7200 USD). Already in April 2021, it exceeded 61,500 USD (according to coinmarketcap.com, accessed: 21 October 2021). We observe a similar enormous growth in the prices of other cryptocurrencies too. Although the prices had fallen at the end of Spring 2021, at the moment of writing this article, they still surpassed the beginning of the 2020 level).

We analysed the returns based on the closing prices, volatility approximated by Garman–Klass estimator [2] and liquidity approximated by the closing quoted spread of Chung and Zhang [3]. We calculated the amount of mutual information contained in the returns of the cryptocurrencies, their volatility and liquidity. We also examined the information transfer between them, both in the pre-COVID-19 and within the COVID-19 period. Our results are validated using the modified DTW algorithm.

Our contributions are threefold. First, we concentrate not only on the volatility but also on the liquidity of the cryptos. The former is no less important from the investors’ perspective during the portfolio selection process. Secondly, we find that the amount of mutual information included in returns and volatility is much higher than the one in liquidity. The latter seems to affect the lead-lag relationships—they are indistinguishable in daily returns and volatility but relatively clear in liquidity. The amount of mutual information contained in liquidity has increased beginning from the pandemic. Moreover, there is no definite leader among cryptocurrencies when it comes to information transfer. We observe the growing role of Ripple in this process, and we link it to the fast transaction processing algorithm of this coin. According to the DTW results, Bitcoin leads all cryptocurrencies in terms of liquidity, but we observe that Monero is its close follower (probably due to the growing interest in more privacy-oriented cryptocurrencies).

Through the study, we enrich our understanding of the information transmission mechanisms in the cryptocurrency market. We also provide some practical information for market participants about the possible benefits of portfolio diversification. Thus, our results are of special importance for the investors. Investment strategies (in any cryptocurrency and not necessarily in Bitcoin) should depend on the observation of prices of a set of cryptocurrencies and not only the most popular one.

## 2. Literature Review

When measuring the dominance of one financial instrument (or market) over another, the most common approach is to investigate the contagion in returns or volatility. With respect to cryptocurrencies, Yi et al. (2018) [1] analysed whether Bitcoin was a dominant cryptocurrency over the period December 2016–April 2018. They found that cryptocurrencies with high market capitalization (namely Bitcoin, Litecoin and Dogecoin) propagate large volatility shocks, while small-cap cryptocurrencies are more likely to receive volatility shocks from others. Although Bitcoin plays an important role and generates strong volatility shocks to other cryptocurrencies, it does not play a role of the ‘clear’ leader on the market in terms of volatility connectedness.

In a similar vein, Ji et al. (2019) [4] applied the measures developed by [5] and found that the return shocks arising from Bitcoin and Litecoin had the most profound effect on the returns of other four large cryptocurrencies between 2015 and 2018. XRM and Ether mostly reacted to negative shocks, while Dash and Ether were weakly reacting to positive returns. In terms of volatility spillovers, Bitcoin was the most powerful and was followed by Litecoin. Ciaian et al. [6] reinforced the conclusion of the lack of the dominant position of Bitcoin. They show that the changes of prices of alternative coins (so-called altcoins) are driven by the development of Bitcoin in the short-run (for 15 out of 16 examined altcoins) but not in the longer term (for only four altcoins).

More closely related work to ours is [7], as that study aimed to detect the informational leadership among four cryptocurrencies, Bitcoin, Ether, Litecoin and XRM. The authors showed that the relationships between cryptocurrencies are nonlinear. Therefore, one should not apply the Granger causality or similar tools that assume linear dependencies in investigating interrelationships among such assets. The scholars utilize a method quite common in econophysics, that is, the group transfer entropy. Their findings indicate that Bitcoin is not a dominating cryptocurrency—it does not lead the information process.

In a more recent paper, Aslanidis et al. [8] documented that the cryptocurrency market experienced a strong overall increase in the connectedness both in terms of returns and volatility. In most cases over the period 2015–2020, shocks were transmitted to the other cryptocurrencies and had a short-term effect on the returns. The scholars also found evidence that the volatility transmission in the high-frequency domain becomes more important than in the low-frequency one. By analysing samples year by year, they found that the variance explained by the first principal component increased over the period both for returns and for volatility. Although over the year ending in August 2016, the percentage of variance explained by the first PC amounted 76% for Bitcoin (the values for Litecoin and Ether were 68% and 7%, respectively); in the year ending in July 2020, the first PC represented 86% of the Bitcoin variance, and the latter was exceeded by Litecoin (91%) and Ethereum (93%). Thus, Bitcoin seems to lose its superior position over time.

The goal of our analysis is to verify whether we can distinguish a leading cryptocurrency. In other words, we are interested on whether cryptocurrencies followed Bitcoin (and we observed causality) during the pandemic and before or the simultaneous increase in the prices of cryptocurrencies reflected the phenomenon of co-occurrence.

## 3. Data

We analyse daily closing prices of the following cryptocurrencies: Bitcoin (BTC), Ether (ETH), Ripple’s XRP, Dash (DSH), Litecoin (LTC), Monero (XMR) and Iota (IOT). These cryptocurrencies vary in terms of the speed of transaction processing, privacy orientation and usage. In the investigated set, Dash and Monero are the leading privacy-oriented cryptos, while Ripple’s XRP processes transactions the fastest. As literature concerning Bitcoin is already saturated [1,4,9], we focus here on the potential successors.

**Ether** is probably the biggest competitor of Bitcoin. At the moment of writing this article, Ether was the second-largest virtual currency by market capitalization in the world. The name Ether refers to the token (or ‘coin’) used through the Ethereum network, launched in 2015. Ether is a medium of exchange similarly to other cryptocurrencies. What sets them apart is that Ether tokens can be used only for one specific purpose: to facilitate the computation of decentralized applications on the Ethereum network [10]. It is possible to exchange different cryptocurrencies for Ether tokens. However, the latter cannot be substituted with other cryptocurrencies to provide computing power for Ethereum transactions.

**XRP** launched in 2012, and it is a cryptocurrency for products developed by Ripple Labs, and that is why these two names, XRP and Ripple, are often used interchangeably. One can use XRP coins for payment settlements, asset exchange and remittance systems. The network RippleNet is utilized by some major banks and financial institutions, e.g., Santander or American Express (see: https://www.ig.com/en/cryptocurrency-trading/cryptocurrency-comparison for details; accessed: 21 October 2021). XRP itself is pre-mined. It uses a less complicated mining method than Bitcoin, which makes the transactions much faster and of a much lower cost [11]. In July 2021, XRP was ranked sixth in terms of total market capitalization (according to coinmarketcap.com).

**Dash** was launched in 2014 and designed to ensure users’ privacy and anonymity. Currently, Dash aims to become a medium for daily transactions, i.e., a digital currency that can be used as cash, credit card or via PayPal [12]. The main difference between Dash and Bitcoin lies in the algorithms applied to mining coins. They also have a different system of validating transactions: In the case of Bitcoin, all the nodes within a network need to validate the transaction, while Dash relies on a specific set of nodes called masternodes. The latter feature enables it to speed up the transaction process [13].

**Litecoin** was founded in 2011 by Christopher Lee. It is called the silver to Bitcoin gold. Its infrastructure is very similar to Bitcoin’s (although the transaction processing speed is faster), so it was used as a test-net for improvements that later were applied to Bitcoin [14]. The limit of LTC coins is 84 Million (than compared to 21 Million of Bitcoin). According to coinmarketcap.com, Litecoin ranked fourteenth in terms of market capitalization in July 2021.

**Monero** is known as the most privacy-oriented cryptocurrency. It was launched in 2014, and its popularity stems from its anonymity orientation [15]. The capitalization of Monero in July 2021 made it 27th among the cryptocurrencies—refer to coinmarkedcap.com.

**IOTA** was launched in 2016. The acronym stands for Internet of Things Application. IOTA is a distributed ledger that handles transactions between connected devices in the IoT. Its cryptocurrency is known as mIOTA [16]. mIOTQ is pre-mined. The method of confirming a transaction results is based on the Tangle infrastructure, with no fees and low power consumption.

Figure 1 presents the volume of trade of the analysed cryptocurrencies, while Figure 2 shows their closing prices. Both prices and volumes are from the Bitfinex exchange. However, as [17] demonstrated, all the crypto-exchanges are very closely linked one to another, and information spills over them almost immediately. Therefore, we can assume that Bitfinex, which has the highest volume of USDBTC trade, can be representative of the market.

What we observe is the peak of volume in each cryptocurrency in March 2020. In the case of BTC, ETH and XMR, that peak is also the maximum observed in the entire analysed period. For the rest of the cryptocurrencies, the maximums of volume traded were present in 2021 (see Figure 1). However, when we compare this picture with Figure 2, we notice that this March 2020 peak was followed by a price drop. Nevertheless, the prices of all cryptocurrencies started to grow steadily, reaching their maximums in 2021.

In Table 1, we provide descriptive statistics of the returns of the analysed cryptocurrencies: mean, standard deviation and kurtosis in two subperiods. The table is accompanied by Figure 3. For mean and standard deviation, we also provide the results of the tests for the equality of the two moments in the analysed subperiods. We conclude that the means were equal in both periods, but standard deviations increased during the pandemic. That is especially visible in Figure 3—we observe an erratic behaviour of returns following the March 2020 price drop. Eventually, we note an interesting phenomenon considering kurtosis. It grew for all the coins, except for Dash. Thus, almost all cryptocurrencies experienced more cases of extreme returns during the pandemic than before it.

## 4. Methods

### 4.1. Volatility and Liquidity Measures

There are various methods to approximate liquidity and volatility. Based on the results presented in [18], we decided to use a Garman–Klass estimator to approximate volatility and the closing quoted spread of Chung and Zhang [3] to approximate liquidity of each cryptocurrency. Both measures require only daily prices; for the Garman and Klass estimator, high-low-open-close prices are employed, while in the case of the closing quoted spread, bid and ask prices are utilized.

To obtain the measures, we used the following formulas: The Garman–Klass [2] volatility estimator:
(1)$$GK=\sqrt{0.5{\left[\log,\left(\frac{H_{t}}{L_{t}}\right)\right]}^{2}-\left(2\log\left(2\right)-1\right)\times {\left[\log,\left(\frac{C_{t}}{O_{t}}\right)\right]}^{2}}$$
where $H_{t}$, $L_{t}$, $O_{t}$ and $C_{t}$ are the high, low, open and close prices in day *t*, respectively.The closing quoted spread of [3]:
(2)$$CQS_{t}=\frac{A_{t}-B_{t}}{0.5(A_{t}+B_{t})}$$
where $B_{t}$ and $A_{t}$ are the bid and the ask prices, respectively, at the end of the given day *t*.

We calculate both the Garman–Klass estimator and the closing quoted spread for each day and each cryptocurrency.

### 4.2. Mutual Information

#### 4.2.1. The Entropy

Mutual information measures the information of a random variable contained in another random variable [19]. It is based on the concept of entropy—i.e., the measure of the uncertainty associated with a random variable (so called Shannon or information entropy [20]).

Let us denote by *X* and *Y* two random variables and assume that each of them can be described by their probability distributions ($P_{X}$ and $P_{Y}$, respectively). The self-information of measuring *X* as outcome *x* is defined as follows [21]:(3)$$I_{X}\left(x\right)=-\log_{2}\left(P_{X}\left(X=x\right)\right)=\log_{2}\frac{1}{P_{X}\left(X=x\right)}.$$

According to [20], for a discrete random variable *X* with probability distribution $P_{X}$, the average number of bits required to optimally encode independent draws can be calculated as follows:(4)$$H_{X}\left(X\right)=-\sum_{x}P_{X}\left(X=x\right)\log_{2}P_{X}\left(X=x\right)=E\left[I_{X}\left(x\right)\right],$$
where $p_{X}\left(x\right)$ denotes a probability density function. The choice of the logarithm’s base only impacts the unit of measurement. Base logarithm indicates bits and base digits, and the base of the natural logarithm yields nats [21].

If we denote the joint distribution of *X* and *Y* by $p_{XY}$, then we can define the joint entropy by the following:(5)$$H\left(X,Y\right)=-\sum_{x}\sum_{y}P_{X,Y}\left(X=x,Y=y\right)\log_{2}\left(P_{X,Y}\left(X=x,Y=y\right)\right)$$

Based on the two measures, one can define conditional entropy as follows:(6)$$H\left(Y|X\right)=H(X,Y)-H(X)$$
and analogously $H\left(X|Y\right)$.

#### 4.2.2. Mutual Information and Global Correlation

Based on the concept of entropy and self-information, one can define mutual information as the following: (7)$$I(X,Y)=H(X)-H\left(X|Y\right)=H(Y)-H\left(Y|X\right)=H(X)+H(Y)-H(X,Y).$$

Mutual information measures the reduction in uncertainty about variable *X* from observing variable *Y*. We will denote it by I(X,Y). Mutual information is positive I(X;Y)≥0. It is equal to 0 if and only if *X* and *Y* are independent.

It is important that mutual information does not imply causality. To account for such a feature, one would need to use transfer entropy (see Section 4.3).

In order to normalize mutual information to take values from 0 to 1 (and be an alternative measure to linear correlation coefficient), Ref. [19] suggested to transform it to the so called global correlation coefficient λ: (8)$$\lambda \left(X,Y\right)=\sqrt{1-\exp\left(-2I(X,Y)\right)}.$$

The function λ(X,Y) captures the overall dependence: both linear and non-linear between *X* and *Y*. It can be interpreted as predictability of *Y* by *X*, where the measure of predictability is based on empirical probability distributions and is model-independent.

### 4.3. Transfer Entropy

Let us assume *X* and *Y* are Markov processes of order *k* and *l*, respectively. Thus, the probability to observe *X* at time t+1 in state *s* conditional on the *k* previous observations is as follows: (9)$$P_{X}\left(X_{t+1}=s|x_{t},\ldots ,x_{t-k+1}\right)=P_{X}\left(X_{t+1}=s|x_{t},\ldots ,x_{t-k}\right).$$

The average number of bits needed to encode the observation in the moment t+1, once the previous *k* values are known, is given by the following: (10)$$h_{X}\left(k\right)=-\sum_{x}P_{X}\left(X_{t+1}=s,x_{t}^{\left(k\right)}\right)\log_{2}P_{X}\left(X_{t+1}=s|x_{t}^{\left(k\right)}\right),$$
where $x_{t}^{\left(k\right)}=x_{t},\ldots ,x_{t-k+1}$.

The information flow from process *Y* to process *X* is measured by quantifying the deviations from the generalized Markov property:$$P_{X}\left(X_{t+1}=s|x_{t}^{\left(k\right)}\right)=P_{X}\left(X_{t+1}=s|x_{t}^{\left(k\right)},y_{t}^{\left(l\right)}\right).$$

The Shannnon transfer entropy measures the information flow from *Y* to *X* and is calculated as the following: (11)$$T_{Y\to X}\left(k,l\right)=\sum P_{X}\left(X_{t+1}=s,x_{t}^{\left(k\right)},y_{t}^{\left(l\right)}\right)\log_{2}\frac{P_{X}\left(X_{t+1}=s|x_{t}^{\left(k\right)},y_{t}^{\left(l\right)}\right)}{P_{X}\left(X_{t+1}=s|x_{t}^{\left(k\right)}\right)}.$$

To calculate the dominant direction of the information flow, one calculates the difference between $T_{Y\to X}$ and $T_{X\to Y}$.

Transfer entropy can also be based on Rényi entropy and is described as follows:(12)$$H_{X}^{q}\left(X\right)=\frac{1}{1-q}\log_{2}\sum_{x}P_{X}^{q}\left(X=x\right).$$

It strongly depends on a weighting parameter q:q>0. For q→1, Rényi entropy converges to Shannon entropy. If we take 0<q<1, then events of a low probability will receive more weight. For q>1, the weights favor outcomes with higher initial probabilities (for further details see: [21]). In the case of financial time series, important information comes in tails. Thus, the authors recommend using small values of *q* and to give more weight to extreme events.

The transfer entropy estimators are biased in small samples. To overcome this problem, one can use the effective entropy measure. It allows for correcting the bias [22]. The effective transfer entropy is defined as follows:(13)$$ET_{Y\to X}\left(k,l\right)=T_{Y\to X}\left(k,l\right)-T_{Y_{sh}\to X}\left(k,l\right)$$
where $T_{Y_{sh}\to X}\left(k,l\right)$ indicates transfer entropy and is calculated using a *shuffled* version of the time series *Y*. This means that the values from the observed time series *Y* are drawn randomly, and they are realigned to generate a new time series.

Rényi transfer entropy is calculated as [21] the following: (14)$$RT_{Y\to X}=\frac{1}{1-q}\log_{2}\frac{\sum_{x}\phi_{q}\left(x_{t}^{\left(k\right)}\right)P_{X}\left(x_{t+1}|x_{t}^{\left(k\right)}\right)}{\sum_{x,y}\phi_{q}\left(x_{t}^{\left(k\right)},y_{t}^{\left(l\right)}\right)P_{X}\left(x_{t+1}|x_{t}^{\left(k\right)},y_{t}^{\left(l\right)}\right)},$$
where the following is called *escort distribution* [21,23]: (15)$$\phi_{q}\left(x\right)=\frac{P_{X}^{q}\left(x\right)}{\sum_{x}P_{X}^{q}\left(x\right)}.$$

If the values of Rényi transfer entropy are negative, then this means that the history of *Y* results in even greater uncertainty than only knowing the history of *X* alone [21].

We calculated transfer entropy and effective transfer entropy by using R package RTransferEntropy [21], and mutual information measure using Infotheo [24].

### 4.4. Dynamic Time Warping

Dynamic Time Warping (futher DTW) is an algorithm used for measuring similarity between two temporal sequences. The goal of the algorithm is to find an optimal alignment between two time series. By optimal alignment, we understand that it achieves the minimum global cost (distance) while ensuring time continuity. The global cost is the summation of the cost between each pair of points in the alignment. The algorithm was first used in speech recognition, where the same signals may differ in speed. It allows for a non-linear mapping of one signal to another by minimizing the distance between the two. The algorithm, unlike econometric methods, does not assume a single delay in the entire period; time series may have different delays at different times. It tries to find the smallest distance among different lags.

Let us assume that we want to compare two time series: a test/query $X=(x_{1},x_{2},\ldots ,x_{N})$ of the length *N* and a reference $Y=(y_{1},y_{2},\ldots ,y_{M})$ of length *M*. We choose a non-negative, local dissimilarity function *f* between any pair of elements $x_{i}$ and $y_{j}$: (16)$$d\left(i,j\right)=f\left(x_{i},y_{j}\right)\geq 0$$
where d(i,j) is small (i.e., low cost) if $x_{i}$ and $y_{j}$ are similar to each other, otherwise d(i,j) is large (i.e., high cost). When employing one of the distance measure (most common Euclidean or Manhattan), the local cost measures for each pair of elements of the sequences *X* and *Y* are evaluated and presented in a cost matrix $C\in R^{N+M}$. A warping path ϕ is a contiguous set of matrix elements that defines a mapping between the time indices of *X* and *Y* that satisfies the boundary, monotonicy and continuity conditions. Given ϕ, the total cost $d_{\phi }$ and the average normalized accumulated cost $\bar{d_{\phi }}$ between the warped time series *X* and *Y* is computed as follows: (17)$$d_{\phi }\left(X,Y\right)=\sum_{k=1}^{T}d\left(\phi_{k}\right),$$
(18)$$\bar{d_{\phi }}\left(X,Y\right)=\sum_{k=1}^{T}\frac{d\left(\phi_{k}\right)m_{\phi }}{M_{\phi }},$$
where $m_{\phi }$ is a per-step weighting coefficient and $M_{\phi }$ is the corresponding normalization constant. The goal is to find an alignment between *X* and *Y* having a minimal average accumulated cost: (19)$$DTW\left(X,Y\right)=min_{\phi }\left\{d_{\phi }\right\}.$$

The optimal path is computed in the reverse order of the indices, starting with (N,M).

In this study, we used extension for this algorithm that is proposed in [25] to check if one time series is forward or backward against the other. We calculated separate DTW distances with windows proposed in [25], finding an optimal path only in the upper triangular cost matrix, within different but always forward shift (called further forward distance $d_{f}$), and in the lower triangular cost matrix, within backward shift ($d_{b}$).

Let us denote two analysed time series by *A* and *B*, the distance measured between each element of *A* and the lagged value of *B* by $d_{f}$ and the distance between the lagged value of *A* and each element of *B* by $d_{b}$. If the distance $d_{f}<d_{b}$, the alignment according to the forward DTW is better, and we call *A* the ‘lead’.

## 5. Results

### 5.1. Amount of Information Shared by the Cryptocurrencies

In Table 2, we present bootstrapped values of the 95% confidence intervals of global correlation coefficients calculated for each pair of the cryptocurrencies. The coefficient measures the amount of the mutual information shared by the returns of each pair of cryptocurrencies. In general, the correlations are high. We observe that in the pre-COVID period, the highest amount of mutual information was shared between XRP and ETH (95% confidence interval of (0.81, 0.86)), BTC and ETH (0.8, 0.86) and LTC and ETH (0.78, 0.85). The pairs XMR and IOT (0.64, 0.76); BTC and IOT (0.66, 0.77); and DSH and IOT (0.66, 0.78) held the lowest amounts of mutual information. The numbers in the lower panel of Table 2 refer to the COVID-19 period. We observe a decline in the value of mutual information shared by the pairs BTC-XMR (0.65, 0.76), ETH-XRP (0.72, 0.8), EHT-XMR (0.66, 0.76) and XRP-XMR (0.63, 0.72), while an increase was observed for DSH-LTC (0.79, 0.86) and DSH-IOT (0.74, 0.82). During this period, the highest amount of information was shared by ETH and LTC (0.8, 0.87), DSH and LTC (0.79, 0.86) and BTC and LTC (0.77, 0.84), whereas the BTC-ETH pair took fourth place (0.77, 0.83).

In Table 3, we present 95% confidence intervals of the global correlation coefficient calculated for volatility. In the pre-COVID-19 period, we observed the highest value of λ for each pair where Ether was present: from (0.66, 0.75) for the pair ETH-XMR to (0.73, 0.81) for ETH-LTC. The values in the pandemics were slightly higher. The only decrease was for the pair ETH-XRP and ETH-IOT. However, we observed the highest increase in linkages for all pairs where LTC, DSH, XMR and IOT were included. This indicates the increase in the importance of these altcoins.

Eventually, in Table 4, we present the analogous calculations for the liquidity of the cryptocurrencies approximated by the CQS measure. In this case, we observe significant growth of relationships. The values of mutual information shared by the liquidity of the cryptocurrencies were rather low before the pandemic. The 95% confidence intervals ranged from (0.24, 0.4) for XRP-BTC to (0.43, 0.57) for IOT-DSH. In the pandemic period, the respective intervals were (0.44, 0.56) for BTC-IOT and (0.6, 0.71) for IOT-DSH. We observed that the leading pair did not change between the periods, but the amount of mutual information shared by it grew. The increase in mutual information shared by liquidity may indicate the overall growth of interest in cryptocurrency trade.

### 5.2. Information Flow between Cryptocurrencies

The analysis of mutual information shared by the cryptocurrencies allows us to conclude that they are strongly interrelated concerning prices and volatility and less interrelated when concerning liquidity. In this section, we verify whether the relationships result in causality. We will concentrate on the causality to and from Bitcoin in the two periods.

In Figure 4, we present the point values of entropy transfer together with their 95% confidence intervals. If the interval covers 0, we conclude that the amount of information transferred is insignificantly different from 0. The calculated entropy was the Renyi one, with q=0.1, i.e., stressing the information in tails. The estimates are each time ordered by the amount of the information flow from BTC.

We observe that the causality relationships between Bitcoin and the other cryptocurrencies are in most cases insignificant, regardless of the medium of interest (returns, volatilities or liquidity). In the pre-COVID-19 period, the information from BTC flew through returns to XMR only. That changed in the COVID-period when the amount of information transmitted to XMR became insignificant. Moreover, the value of the transfer entropy became negative when we analysed the direction from BTC to XRP.

When it comes to volatility and liquidity, we observed no significantly positive information transfer from BTC in any period. On the contrary, in the pre-COVID one, the values of the transfer entropy were negative for the information transfer from BTC to XMR through volatility and liquidity and to IOT through liquidity. In the COVID period, all the values became insignificantly different from 0.

When it comes to information transfer to BTC through returns, we noted positive values in the pre-COVID-19 period for XMR and in the COVID-19 one for LTC only. The transfer entropy from DSH was significantly negative. We also emphasize that some negative values of transfer entropy observed in volatility (transfer from XMR and LTC) became insignificant in the second period. The fact that some values of transfer entropy were negative implies that any investment strategy based on inferring the returns or volatility of Bitcoin based on the historical returns or volatility dynamics of any other from our set may be ineffective.

Eventually, when we concentrate on liquidity, we observe that, during the pandemic, the values of transfer entropy were the highest in the case of XRP (the 95% interval limit still covers 0, but the part of the interval taking negative values is the shortest among all the cryptocurrencies). We explain this result by the fact that XRP is characterized by the fastest transaction processing algorithm and has the potential to lead the information process in liquidity. Moreover, interest in this cryptocurrency is steadily growing.

By summarizing the results and comparing the values of the transfer entropy to and from Bitcoin, we can say that in terms of returns, BTC is the information receiver and that returns dynamic from smaller coins influence the dynamics of the big one more than the other way round.

### 5.3. Lead-Lag Relationships

In order to extend the results obtained by analysing transfer entropy, we also calculated the lead-lag relationships between the cryptocurrencies using the DTW algorithm. When analysing the transfer entropy, we allowed for one lag only, while in the DTW algorithm, we took into account the 7-day history.

In Table 5, Table 6 and Table 7, we present the differences between the forward and the backward distances for returns, volatilities and liquidities, respectively. The negative values in Table 5, Table 6 and Table 7 denote that we can treat the currency in the column as the leading one relative to the one from the row. The table can be read in two directions so that the positive value shows that the currency in the row can be read as a leader. The gray colour indicates that the currency from the column switched its role from leader to follower compared to the pre-pandemic period. The orange colour signifies the change in the opposite direction.

To render the information in the tables clearer, let us concentrate on the relationship in volatility between BTC and ETH from 1 January 2019 to 1 March 2020. The forward DTW distance of the BTC to ETH takes into account the alignment of the current volatility of BTC with the future volatility of ETH, accounting for different shifts from 1 to 7 days. It amounted to 0.0080 (not included in the table). The backward DTW (the current BTC volatility match with past ETH volatility with different lags from 1 to 7 days) amounted to 0.0071. In the Table 6, we display the difference (multiplied by 100) between the two values, which are equal to 0.09. The positive sign means that BTC is the leading currency in this pair. The absolute value of these differences is not high, but we can observe that from 1 March 2020 to 30 June 2021 the value rose to 0.19. The difference has more than doubled; thus, we can conclude that the position of BTC as a volatility-leader against ETH, has strengthened.

We note two facts. First of all, the numbers presented in the tables represent differences between the distances and not the estimates of parameters. Therefore, we do not present here the significance tests. Instead, we can comment on the magnitude of the numbers. All the numbers in the tables are multiplied by 100. The differences between returns and volatilities are very small. That can suggest that daily data were not enough to capture the lead-lag relationships in returns and volatilities. It is likely that such relationships are more pronounced in intra-daily data. On the contrary, the differences between liquidities are relatively high.

The results corroborate with the one obtained by the analysis of mutual information. The highest amount of mutual information is shared by returns and volatilities. This is likely why we observed such small differences between forward and backward distances in Table 5 and Table 6. Since the amount of mutual information contained in liquidities is smaller (see Table 4), clearer lead-lag relationships can be observed.

Let us concentrate on Table 5. In both periods, BTC slightly leads; however, the absolute value of the difference between the forward and backward distances is higher during the pandemic period for the relationship with ETH (0.04 vs. 0), XRP(0.29 vs. 0.05), DSH (0.2 vs. 0.08) and XMR(0.02 vs. 0) but not for IOT (0.19 vs. 0.22) and LTC (0 vs. 0.24). In the pandemic period, ETH is a little ‘lagged’ relative to XRP, BTC but also XMR. The absolute value of differences between the forward and backward distances between LTC and other cryptocurrencies decreased a little during the pandemic.

The results for the volatility are presented in Table 6. In the pandemic period, ETH became a slight leader relative to XRP (−0.34), DSH (−0.03), XMR (−0.01) and IOT (−0.14). LTC volatility became forward relative to ETH (−0.03), XRP (−0.22), XMR (−0.02) and IOT (−0.09), while XRP became a little backward compared to all others.

Eventually, in the case of liquidity analysis (cf. Table 7), we can draw much stronger conclusions as the numbers are much higher. BTC is forward relative to all others cryptocurrencies before as well as during the pandemic. XMR is a leader relative to all others, despite BTC in the pandemic period.

## 6. Discussion and Conclusions

In the article, we present the results of the analysis of mutual information, information transfer and lead-lag relationships between returns, volatility and liquidity of cryptocurrencies. We found that cryptocurrencies share a relatively high amount of mutual information (especially in returns and volatility), while information transfer between them is limited. Moreover, we observed that mutual information shared in liquidity has increased since the beginning of the pandemic. The lead-lag relationships between Bitcoin and other cryptocurrencies in terms of returns and volatility are almost indistinguishable in daily data, which is probably related to the high amount of mutual information shared by these measures. Additionally, using dynamic time warping, we have found that changes in the liquidity of Monero (XMR) started to precede the changes in liquidity of all other cryptocurrencies, apart from Bitcoin.

Our results partially corroborate with the ones presented in the previous studies and obtained with different econometric methods. Similarly to [1,6,7], we show that the dominance of Bitcoin is not definite, although it has been the most recognizable cryptocurrency. Demonstrating the significant information transfer from Litecoin to Bitcoin through returns, we also corroborate the results presented in [4]. We confirm that high-capitalization cryptocurrencies (Bitcoin, Ether and Litecoin) share a large amount of mutual information with others. However, over time, the relationships become weaker. Moreover, it is most visible in returns.

We note that mutual information contained in volatility and returns is higher than the one in liquidity, and the maximal numbers are reached for returns. The latter suggests that although all cryptocurrencies may experience similar price dynamics, the market values their risk differently. We can also infer that the cryptocurrency market is divided into segments with different groups of investors. In general, the investment strategy in alternative coins based on observing Bitcoin seems to be inadequate. Investors should take into account the information flow from other currencies as well.

In future work, we plan to repeat the research for a longer time frame in order to verify the stability of the results in time. Together with extending the period of the study, we aim to include more altcoins in our research. We intend to verify the possibility of hedging the investment in the dominating cryptocurrencies with altcoins in the long run.

## Figures and Tables

**Figure 1 entropy-23-01386-f001:**
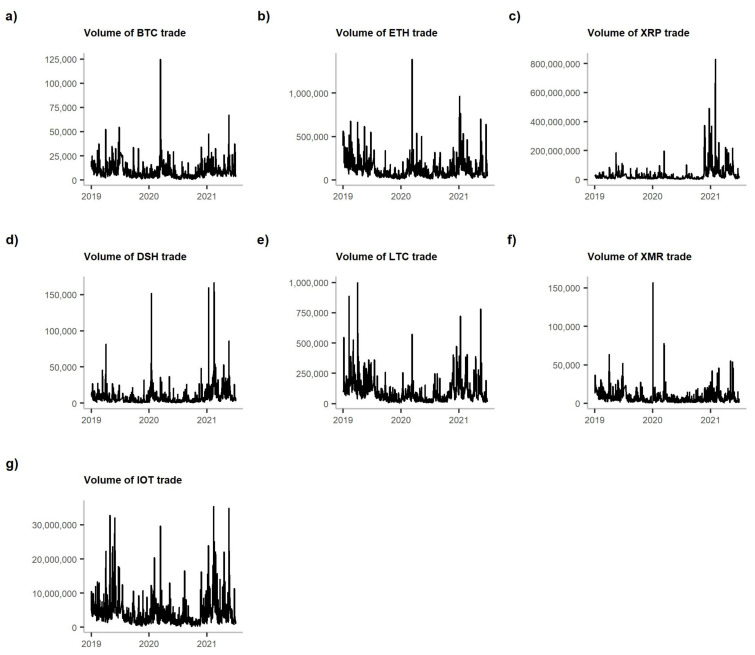
The volume of trade of the analysed cryptocurrencies. **Note:** The graphs are shown in the following order: (**a**) BTC, (**b**) ETH, (**c**) XRP, (**d**) DSH, (**e**) LTC, (**f**) XMR and (**g**) IOT.

**Figure 2 entropy-23-01386-f002:**
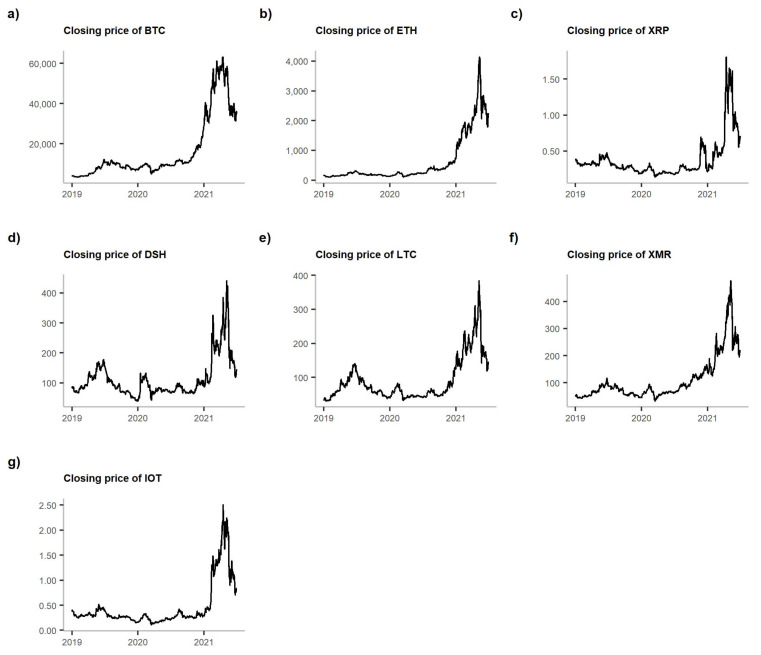
Closing prices (in USD) of the analysed cryptocurrencies. **Note:** The graphs are shown in the following order: (**a**) BTC, (**b**) ETH, (**c**) XRP, (**d**) DSH, (**e**) LTC, (**f**) XMR and (**g**) IOT.

**Figure 3 entropy-23-01386-f003:**
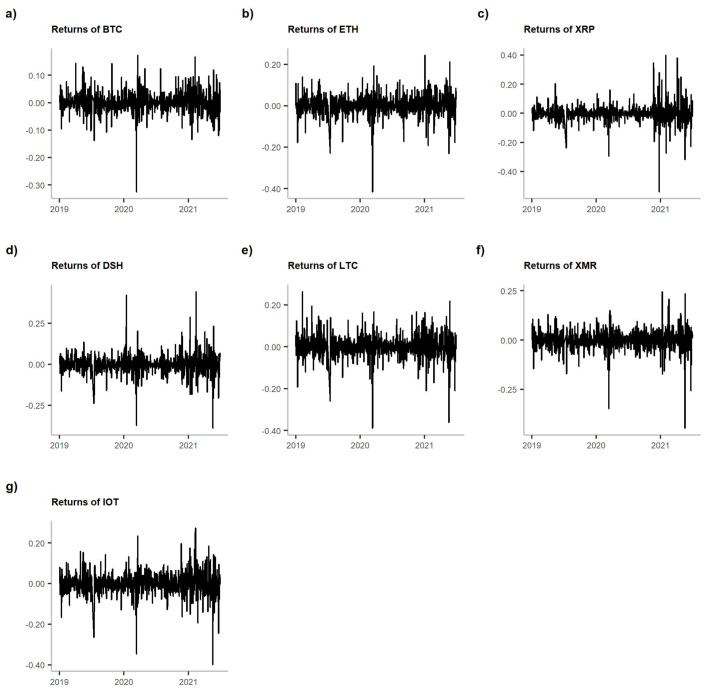
Log-returns of the analysed cryptocurrencies. **Note:** The graphs are shown in the following order: (**a**) BTC, (**b**) ETH, (**c**) XRP, (**d**) DSH, (**e**) LTC, (**f**) XMR and (**g**) IOT.

**Figure 4 entropy-23-01386-f004:**
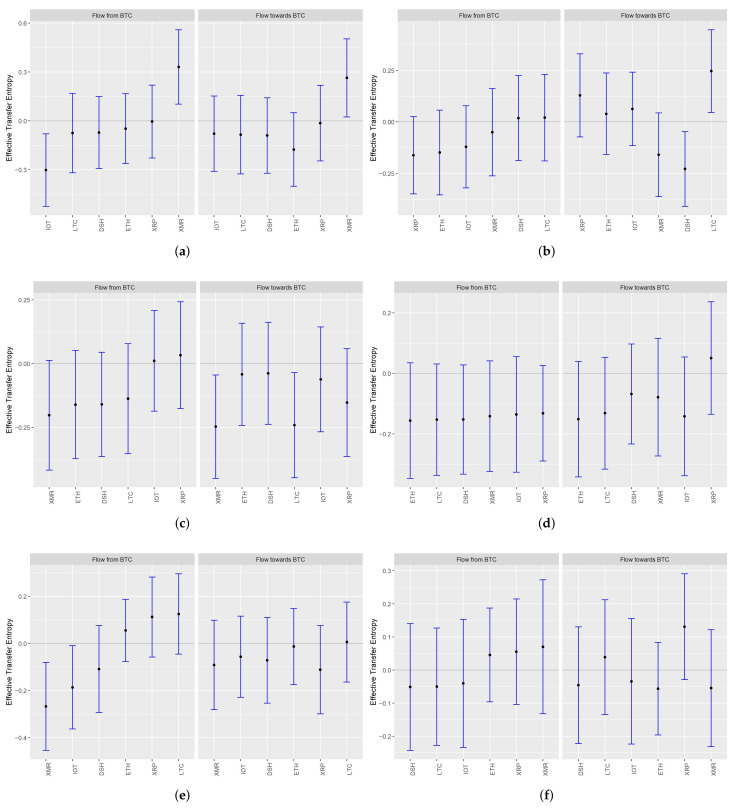
Effective Renyi entropy transfer from Bitcoin (BTC) to other cryptocurrencies and from those to BTC. **Note**: In the figure, we present the values of effective transfer entropy between the returns (**a**,**b**), volatilities (**c**,**d**) and liquidities (**e**,**f**) of cryptocurrencies calculated according to Equation (Equation 14) for q=0.1, and the Markov order is set to 1 for each coin. The point values are accompanied by 95% confidence bands. Intervals covering positive values denote statistically significant causality. The negative ones indicate the increase in uncertainty when accounting for the history of other cryptocurrencies. If the intervals cover 0, we interpret it as a lack of a statistically significant relationship. The left column refers to the pre-COVID-19 period, while the right refers to the COVID-19 one. The currencies on the x-axis are ordered by the growing amount of the transfer entropy. Therefore, the order of the cryptocurrencies differs between the periods. The data have been discretized using quantiles methodology.

**Table 1 entropy-23-01386-t001:** Descriptive statistics of cryptocurrencies’ return series.

Moment	Period	BTC	ETH	XRP	DSH	LTC	XMR	IOT
mean μ	pre-COVID	0.002	0.001	−0.001	0.000	0.001	0.001	−0.001
	COVID	0.003	0.005	0.002	0.001	0.002	0.002	0.003
*p*-val for H0: $\mu_{1}=\mu_{2}$		0.687	0.253	0.377	0.792	0.900	0.580	0.250
st.dev. σ	pre-COVID	0.033	0.043	0.040	0.050	0.051	0.041	0.045
	COVID	0.042	0.056	0.075	0.068	0.059	0.057	0.064
*p*-val for H0: $\sigma_{1}=\sigma_{2}$		<0.01	<0.01	<0.01	<0.01	<0.01	<0.01	<0.01
kurtosis	pre-COVID	3.428	3.910	5.633	14.300	4.313	1.505	4.185
	COVID	8.989	8.281	11.127	8.669	7.598	12.223	6.337

**Note**: μ denotes mean, while σ standard deviation. The data are taken daily.

**Table 2 entropy-23-01386-t002:** Global correlation coefficient for cryptocurrencies’ returns: before and during pandemic.

**From 1 January 2019 to 1 March 2020**						
	**ETH**	**XRP**	**DSH**	**LTC**	**XMR**	**IOT**
BTC	(0.8, 0.86)	(0.71, 0.81)	(0.73, 0.81)	(0.72, 0.82)	(0.76, 0.84)	(0.66, 0.77)
ETH	–	(0.81, 0.86)	(0.74, 0.82)	(0.78, 0.85)	(0.72, 0.81)	(0.71, 0.81)
XRP	–	–	(0.71, 0.79)	(0.74, 0.83)	(0.69, 0.78)	(0.73, 0.82)
DSH	–	–	–	(0.68, 0.79)	(0.7, 0.8)	(0.66, 0.78)
LTC	–	–	–	–	(0.68, 0.78)	(0.69, 0.78)
XMR	–	–	–	–	–	(0.64, 0.76)
**From 1 March 2020 to 30 June 2021**						
BTC	(0.77, 0.83)	(0.68, 0.77)	(0.73, 0.82)	(0.77, 0.84)	(0.65, 0.76)	(0.68, 0.76)
ETH	–	(0.72, 0.8)	(0.75, 0.82)	(0.8, 0.87)	(0.66, 0.76)	(0.72, 0.79)
XRP	–	–	(0.73, 0.81)	(0.74, 0.81)	(0.63, 0.72)	(0.72, 0.79)
DSH	–	–	–	(0.79, 0.86)	(0.67, 0.75)	(0.74, 0.82)
LTC	–	–	–	–	(0.67, 0.77)	(0.73, 0.8)
XMR	–	–	–	–	–	(0.65, 0.74)

**Note**: In the table, we present the bootstrapped 95% confidence intervals of the global correlation coefficient (in nats) calculated according to Equation (Equation 8). Values closer to 1 denote a higher reduction in uncertainty when observing the returns of the cryptocurrency from the row. The global correlation coefficient is a measure of interdependence but not causality. The data have been discretized by using equal frequencies binning algorithm, and the number of bins was set to $\sqrt[{3}]{N}$, where *N* is the sample length.

**Table 3 entropy-23-01386-t003:** Global correlation coefficients for cryptocurrencies volatility approximated by the Garman–Klass [2] estimator.

**From 1 January 2019 to 1 March 2020**						
	**ETH**	**XRP**	**DSH**	**LTC**	**XMR**	**IOT**
BTC	(0.68, 0.78)	(0.61, 0.73)	(0.58, 0.7)	(0.65, 0.75)	(0.68, 0.78)	(0.61, 0.71)
ETH	–	(0.71, 0.79)	(0.67, 0.76)	(0.73, 0.81)	(0.66, 0.75)	(0.7, 0.79)
XRP	–	–	(0.6, 0.72)	(0.65, 0.74)	(0.65, 0.73)	(0.69, 0.77)
DSH	–	–	–	(0.62, 0.73)	(0.62, 0.73)	(0.62, 0.72)
LTC	–	–	–	–	(0.63, 0.73)	(0.64, 0.74)
XMR	–	–	–	–	–	(0.6, 0.71)
**From 1 March 2020 to 30 June 2021**						
BTC	(0.74, 0.81)	(0.67, 0.75)	(0.71, 0.79)	(0.74, 0.8)	(0.72, 0.79)	(0.67, 0.75)
ETH	–	(0.68, 0.75)	(0.71, 0.8)	(0.75, 0.82)	(0.7, 0.78)	(0.7, 0.77)
XRP	–	–	(0.74, 0.81)	(0.79, 0.84)	(0.66, 0.75)	(0.71, 0.79)
DSH	–	–	–	(0.79, 0.85)	(0.74, 0.8)	(0.74, 0.81)
LTC	–	–	–	–	(0.71, 0.79)	(0.73, 0.8)
XMR	–	–	–	–	–	(0.7, 0.79)

**Note**: In the table, we present bootstrapped 95% confidence intervals of the global correlation coefficient (in nats) calculated according to Equation (Equation 8). Values closer to 1 denote a higher reduction in uncertainty when observing the volatility—Equation (Equation 1)—of the cryptocurrency from the row. The global correlation coefficient is a measure of interdependence but not causality. The data have been discretized by using equal frequencies binning algorithm, and the number of bins was set to $\sqrt[{3}]{N}$, where *N* is the sample length.

**Table 4 entropy-23-01386-t004:** Global correlation coefficient for cryptocurrencies liquidity approximated by the closing quoted spread of [3] (CQS).

**From 1 January 2019 to 1 March 2020**						
	**ETH**	**XRP**	**DSH**	**LTC**	**XMR**	**IOT**
BTC	(0.31, 0.52)	(0.24, 0.4)	(0.29, 0.47)	(0.3, 0.45)	(0.33, 0.49)	(0.29, 0.45)
ETH	–	(0.3, 0.46)	(0.3, 0.49)	(0.31, 0.5)	(0.3, 0.49)	(0.32, 0.5)
XRP	–	–	(0.33, 0.48)	(0.36, 0.51)	(0.36, 0.51)	(0.35, 0.5)
DSH	–	–	–	(0.38, 0.52)	(0.37, 0.52)	(0.43, 0.57)
LTC	–	–	–	–	(0.32, 0.49)	(0.37, 0.52)
XMR	–	–	–	–	–	(0.36, 0.5)
**From 1 March 2020 to 30 June 2021**						
BTC	(0.52, 0.64)	(0.5, 0.61)	(0.4, 0.54)	(0.51, 0.63)	(0.49, 0.62)	(0.44, 0.56)
ETH	–	(0.58, 0.69)	(0.53, 0.65)	(0.57, 0.69)	(0.59, 0.69)	(0.54, 0.65)
XRP	–	–	(0.55, 0.67)	(0.59, 0.69)	(0.56, 0.67)	(0.53, 0.66)
DSH	–	–	–	(0.54, 0.66)	(0.56, 0.67)	(0.6, 0.71)
LTC	–	–	–	–	(0.52, 0.63)	(0.5, 0.62)
XMR	–	–	–	–	–	(0.59, 0.7)

**Note**: In the table, we present bootstrapped 95% confidence intervals of the global correlation coefficient (in nats) calculated according to Equation (Equation 8). Values closer to 1 denote a higher reduction in uncertainty when observing the liquidity—Equation (Equation 2)—of the cryptocurrency from the row. The global correlation coefficient is a measure of interdependence but not causality.The data have been discretized using equal frequencies binning algorithm, and the number of bins was set to $\sqrt[{3}]{N}$, where *N* is the sample length.

**Table 5 entropy-23-01386-t005:** Differences between forward and backward DTW measures of the returns.

	ETH	XRP	DSH	LTC	XMR	IOT
from 1 January 2019 to 1 March 2020						
BTC	0.00	0.05	0.08	0.24	0.00	0.22
ETH		−0.06	0.05	0.01	0.02	0.00
XRP			0.00	0.14	0.04	−0.01
DSH				0.04	0.00	0.05
LTC					−0.10	−0.05
XMR						0.00
from 1 March 2020 to 30 June 2021						
BTC	0.04	0.29	0.20	0.00	0.02	0.19
ETH		−0.01	0.02	0.00	−0.03	0.03
XRP			0.05	0.02	−0.03	0.00
DSH				0.05	0.01	0.06
LTC					0.03	0.01
XMR						−0.01

**Note**: In the table, we present the differences between forward and backward distances (multiplied by 100 for clarity). The negative values denote that we can treat the currency in the column as the leading one relative to the one from the row. Gray colour indicates the switch of the relationship from leaders to followers during the pandemic compared to the pre-pandemic period, while orange means changes in the opposite direction.

**Table 6 entropy-23-01386-t006:** Differences of forward and backward DTW distances of cryptocurrencies’volatility.

	ETH	XRP	DSH	LTC	XMR	IOT
From 1 January 2019 to 1 March 2020						
BTC	0.09	0.10	0.17	0.26	0.00	0.23
ETH		−0.05	−0.03	0.14	−0.05	0.00
XRP			0.01	0.16	0.02	0.10
DSH				0.13	−0.05	0.04
LTC					−0.18	−0.09
XMR						0.09
From 1 March 2020 to 30 June 2021						
BTC	0.19	0.66	0.18	0.18	0.13	0.30
ETH		0.34	0.03	−0.03	0.01	0.14
XRP			−0.18	−0.22	−0.34	−0.07
DSH				0.01	0.00	0.00
LTC					0.02	0.09
XMR						0.15

**Note**: In the table we present the differences between forward and backward distances (multiplied by 100 for clarity). The negative values denote that we can treat the currency in the column as the leading one relative to the one from the row. Gray colour indicates the switch of the relationship from leaders to followers during the pandemic compared to the pre-pandemic period, while orange means changes in the opposite direction.

**Table 7 entropy-23-01386-t007:** Differences of forward and backward DTW distances of the cryptocurrencies’ liquidity.

	ETH	XRP	DSH	LTC	XMR	IOT
From 1 January 2019 to 1 March 2020						
BTC	43.89	47.15	7.19	51.93	10.14	52.94
ETH		61.13	4.17	17.14	−0.17	66.92
XRP			69.65	36.22	117.58	−2.78
DSH				20.91	2.52	−64.31
LTC					−19.74	−34.53
XMR						−112.67
From 1 March 2020 to 30 June 2021						
BTC	83.81	121.13	60.70	78.80	52.06	106.64
ETH		95.25	−10.60	6.26	−8.90	80.76
XRP			8.65	−39.52	−39.50	−6.00
DSH				13.30	−0.49	−23.75
LTC					−15.17	20.61
XMR						24.05

**Note**: In the table we present the differences between forward and backward distances (multiplied by 100 for clarity). The negative values denote that we can treat the currency in the column as the leading one relative to the one from the row. Gray colour indicates the switch of the relationship from leaders to followers during the pandemic compared to the pre-pandemic period, while orange means changes in the opposite direction.

## Data Availability

All the data comes from BitFinex exchange and were obtained through QUANDL.

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
