# Peer review of "Is Bitcoin Still a King? Relationships between Prices, Volatility and Liquidity of Cryptocurrencies during the Pandemic"

_entropy, 2021, doi:10.3390/e23111386_

Round 1

Reviewer 1 Report

(1) The various entropy measures reported depend on the choice of "bins" used to discretize probability distributions. Without knowing the choice of bins, it is difficult to interpret the results. The authors should be clear about the choice of bins. Relatedly, there is no need to discuss the continuous case in Section 4.2.1 if (as I presume) everything is discretized.

(2) The various correlations and entropies reported in Tables 2-9 are not accompanied by any reporting of statistical reliability, e.g. p-values. Such information should be provided (and can easily be obtained from the cited R package RTransferEntropy, for instance).

(3) For completeness, a p-value for kurtosis stability should be included in Table 1.

Reviewer 2 Report

Summary

This study investigate the lead-lag relationship between cryptocurrencies during the period between January 2019 and June 2021. Specifically, this study applies transfer entropy, effective transfer entropy, and mutual information measure to examine the lead-lag relationship between cryptocurrencies. Measures of return, volatility, and liquidity are examined. Overall, this is an interesting study. My comments are in the followings.

(1) As shown in Table 2, the price movements (returns) are highly correlated between each other for the 7 selected cryptocurrencies. Although there is a little decrease during the pandemic period, I do not think the difference can be significant from zero. Therefore, comparing the differences of the relationship (or the lead-lag effect) between crytocurrencies will be a relatively minor issue.

(2) Tables 5 to 7 and Tables 8 to 9 are the main results of this study. However, it is hard to link numbers presented in tables to the conclusion/implications in the manuscript. I would suggest the author explain and discuss the results in a detailed way.

(3) In figures 1 and 2, the author should define (a), (b), (c), (d), (e), (f), and (g).

(4) In terms of return, standard deviation, and kurtosis, what is the data frequency for those variables?

(5) In Figure 2, figures with different levels cannot be compared to each other. In addition, it has no sense to compare absolute price between securities. Using returns or cumulative returns is suggested.

Round 2

Reviewer 1 Report

(1) The p-values reported in Tables 2-4 are all the same: "<0.01". It would be much more informative to replace these p-values with standard errors or confidence bands.

(2) The new captions for Figures 2-4 say "we present the bootstrapped values of the global correlation coefficient", but I don't think this is correct. The correlation coefficients are not bootstrapped. The bootstrap is used to construct p-values (or standard errors, or confidence bands) for the correlation coefficients.

Author Response

We would like to thank the Reviewer for the comments. In the new version we took into account all the remarks. According to the remark (1), we repeated the calculations and now we provide the bootstraped 95% confidence intervals.  We agree with the remark (2), but since the p-values have been removed, the sentence do not appear in the text anymore.